# The Effect of APOE ɛ4 on the Functional Connectivity in Frontoparietal Network in Hypertensive Patients

**DOI:** 10.3390/brainsci12050515

**Published:** 2022-04-19

**Authors:** Dandan Wang, Chang Xu, Wenxiao Wang, Hui Lu, Junying Zhang, Furu Liang, Xin Li

**Affiliations:** 1State Key Laboratory of Cognitive Neuroscience and Learning & IDG/McGovern Institute for Brain Research, Beijing Normal University, Beijing 100875, China; wdandan_no1@163.com (D.W.); b_bbecky@163.com (C.X.); xiaoxiaxbx@163.com (W.W.); luhuinn@163.com (H.L.); 2Beijing Aging Brain Rejuvenation Initiative (BABRI) Center, Beijing Normal University, Beijing 100875, China; 3Institute of Basic Research in Clinical Medicine, China Academy of Traditional Chinese Medicine, Beijing 100700, China; zhangjuny1985@163.com; 4Department of Neurology, Baotou Central Hospital, Baotou 014040, China; ru_liang@sina.com

**Keywords:** hypertension, APOE, alzheimer disease, functional connectivity, resting-state network, executive function

## Abstract

Allele 4 of the apolipoprotein E gene (APOE ε4) and hypertension are considered risk factors for Alzheimer’s Disease (AD). The detection of differences in cognitive function and brain networks between hypertensive patients who are APOE ε4 carriers and non-carriers may help in understanding how hypertension and risk genes cumulatively impair brain function, which could provide critical insights into the genetic mechanism by which hypertension serves as a potential risk factor for cognitive decline and even AD. Using behavioral data from 233 elderly hypertensive patients and neuroimaging data from 38 of them from Beijing, China; the study aimed to assess the effects of APOE ε4 on cognition and to explore related changes in functional connectivity. Cognitively, the patients with APOE ε4 showed decreased executive function, memory and language. In the MRI sub-cohort, the frontoparietal networks in the APOE ε4 carrier group exhibited an altered pattern, mainly in the left precentral regions, inferior frontal lobe and angular gyrus. More importantly, the decline of cognitive function was correlated with abnormal FC in the left precentral regions in APOE ε4 carriers. APOE ε4 aggravated the dysfunction in frontal and parietal regions in hypertensive patients. This highlights the importance of brain protection in hypertensive patients, especially those with a genetic risk of AD.

## 1. Introduction

Hypertension is a well-established risk factor for cognitive impairments and dementia [1,2]. However, not every person with hypertension develops cognitive impairments, suggesting an interaction with other determinants, e.g., genetic factors. The apolipoprotein E (APOE) epsilon 4 (ε4) allele is well established as the strongest genetic risk factor for cognitive impairments [3,4] and hypertension [5], and therefore, may moderate the association between hypertension and cognitive decline. However, the mechanism of how APOE ε4 and hypertension lead to the increased risk of cognitive impairments remains unclear.

The panoply of cognitive functions requires coordination among networked brain regions [6,7]. In the brain, there are some intrinsic functional resting-state networks (RSNs) [8,9], which are related to cognitive impairments. Previous studies have found that there is alternation in the frontoparietal network (especially in the posterior parietal cortex) and salience network (especially in the rostral prefrontal cortex) in hypertensive patients [10,11,12]. The decreased connectivity efficiency in the frontoparietal network and salience network is related to the worse performance of executive functions [13]. Meanwhile, the APOE-ε4 allele is also associated with localized brain functional alterations. Most consistently, studies showed decreased default mode network (DMN) connectivity in parietal and frontal areas [9,10,11,12,13,14,15,16], increased central executive network (CEN) connectivity in parietal areas [14], such as the inferior parietal lobule, and increased connectivity in the salience network (SN) [15,16,17] in older APOE-ε4 carriers. Disrupted connectivity patterns are evident, respectively, in hypertensive patients and the elderly with APOE ε4, suggesting that breakdowns in this interregional choreography lead to dysfunction [7].

Anatomically, the white matter (WM) tracts are considered the anatomic links of RSNs and densely interconnect the regions within these RSNs [18]. Interestingly, both hypertension and APOE ε4 influence the WM tracts, especially in the frontal and subcortical regions [19]. There is a significantly higher subcortical white matter lesion volume in APOE ε4 carriers with hypertension than in non-ε4 carriers [20]. Based on these previous findings, we could infer the existence of the joint effect of the APOE risk gene and hypertension on the functional connectivity (FC) of RSNs. Exploring the FC of RSN networks could help our understanding of the effect of APOEε4 on cognitive function in hypertensive patients.

This study aimed to assess the cognitive performance in hypertensive patients with APOE ε4 in a sample of individuals of Han nationality and the related neural mechanisms. We hypothesized that the hypertension patients with APOE ε4 carrier would be associated with changed functional connectivity of frontoparietal networks, leading to deficits in cognitive functions. The study of the cognitive impairment due to genetic factors in hypertensive patients could help in understanding how hypertension and risk genes cumulatively impair brain function and build an early warning for dementia. To the best of our knowledge, such a study on alterations in RSNs in hypertension is not available in the literature.

## 2. Materials and Methods

### 2.1. Large Sample of Behavioral Research

#### 2.1.1. Participants

The participants in the present study were from the Beijing Aging Brain Rejuvenation Initiative (BABRI) [21]. Participants with addictions, psychiatric diseases, those undergoing treatments that would affect cognitive function, or who were unable to complete the neuropsychological tests were excluded. All of the hypertensive patients were measured with standard laboratory testing by a specialist physician. The blood pressures were controlled below 140/90 mmHg [10]. They all had a history of using oral antihypertensive medications based on their medical records. The antihypertensive medications included angiotensin receptor blockers, calcium channel blockers, diuretics, β blockers, and compound antihypertensives. There were 233 qualified hypertensive participants who completed the neuropsychological tests, personal information questionnaire and genotyping. All participants gave written informed consent to our protocol, which was approved by the ethics committee of the State Key Laboratory of Cognitive Neuroscience and Learning, Beijing Normal University.

#### 2.1.2. Analysis of Genotyping

Participants were pre-screened for the APOE genotype using a TaqMan SNP genotyping assay on a 7900HT Fast Real-Time PCR system (Applied Biosystems, Foster City, CA, USA). DNA was extracted from the blood samples of subjects for the subsequent characterization of the APOE genotype via PCR according to standard procedures. All participants were genotyped for two SNPs in the APOE gene (rs429358 and rs7412) using previously published methods [22,23]. Genotype identifications were manually and independently verified by two laboratory personnel. Ten percent of the sample’s genotypes underwent quality control duplication. Thirty-four APOEε4 carriers and 199 APOE ε4 non-carriers were included in our present study.

#### 2.1.3. Neuropsychological Tests and Personal Information Questionnaire

A group of students who were trained by professional neuropsychologists performed the neuropsychological tests and the personal information questionnaire on these study participants. The Chinese translation of the Mini-Mental State Examination (MMSE) served as a general cognitive function test. The neuropsychological battery included memory (Auditory Verbal Learning Test (AVLT), Rey–Osterrieth Complex Figure test (ROCF) (delay), forward and backward Digit Span), visuospatial ability (ROCF (copy), Clock-Drawing Test (CDT)), language (Category Verbal Fluency Test (CVFT), Boston Naming Test (BNT)), processing speed (Trail Making Test (TMT) A and Stroop Color and Word Test (SCWT)-B) and executive function (TMT-B and SCWT-C-B) cognition domains. The personal information questionnaire included demographic information, such as a series of chronic diseases including hypertension, coronary heart disease, diabetes mellitus, cerebrovascular disease, chronic bronchitis or emphysema, osteoarthritis and intervertebral disk disease and medical history.

### 2.2. MRI Studies

#### 2.2.1. Participants

All Magnetic resonance imaging (MRI) acquisitions were performed no more than one month after the neuropsychological tests. Exclusion criteria included a previous history of chronic disease, neurological disease and unsuitability for MRI (e.g., due to metal dentures, metal prostheses, prosthetic heart valve, stents, pacemakers, claustrophobia, or Meniere’s syndrome diseases). There were 19 qualified hypertensive participants carrying the APOE ε4, and then chose 19 age-, education- and sex-matched non-carriers (Figure 1) from the 233 total subjects to MRI scans.

#### 2.2.2. MRI Data Acquisition

MRI data were acquired using a SIEMENS TRIO 3T scanner in the Imaging Center for Brain Research, Beijing Normal University and included resting-state functional magnetic resonance imaging (rsfMRI) scans [24]. To minimize head movement, participants lay supine with their heads snugly fixed by straps and foam pads. Resting-state data were collected using a gradient echo EPI sequence [TE = 30 ms, TR = 2000 ms, flip angle = 90°, 33 slices, slice thickness = 3.5 mm, matrix = 64 × 64, and field of view (FOV) = 200 × 200 mm^2^, acquisition time = 8 min and 5 s].

#### 2.2.3. Resting-State fMRI Data Analysis

For each participant, the first 10 volumes were discarded to allow the participants to adapt to the magnetic field. Functional data were preprocessed using SPM8 (http://www.fil.ion.ucl.ac.uk/spm) (accessed on 19 September 2019) and DPARSF (http://www.restfmri.net/forum/taxonomy/term/36) (accessed on 3 September 2019), including slice timing, within-subject inter-scan realignment to correct possible movement, spatial normalization to a standard brain template in the Montreal Neurological Institute coordinate space, resampling to 3 × 3 × 3 mm^3^ and smoothing with an 8-mm full-width half-maximum Gaussian kernel. Besides, rsfMRI data were processed with linear detrending, 0.01–0.08 Hz bandpass filtering and regression correction for nuisance covariates, including six motion parameters, the global mean signal, the white matter signal and the cerebrospinal fluid signal.

We performed an independent component analysis (ICA) using the group ICA toolbox (GIFT version 2.0e; http://mialab.mrn.org/software/gift/ accessed on 1 March 2020) with 25 independent components separately estimated for each group. There were three main stages: (i) a principal component analysis was performed for each subject for data reduction, (ii) the ICA algorithm was applied, and (iii) back-reconstruction was performed for each individual subject [23]. The best-fit components for the left frontoparietal (LFP), right frontoparietal (RFP) network, default mode network (DMN) and salience network (SN) were identified by visual inspection. For each network, ANCOVA tests were used to compare FC z values to determine the significance of between-group differences (*p* < 0.05, FWE corrected).

### 2.3. Statistical Analysis

ANCOVA analyses were used to compare neuropsychological assessments between the two groups, with age, education, gender, and medical history (including coronary heart disease, cerebrovascular disease and diabetes) as covariates. A generalized linear model (GLM) with functional connectivity and hypertension with APOE ε4 carriers-status as predictor variables was used and controlled the above covariates. If significant effects exist, Pearson’s partial correlation analysis was continued and performed separately for the APOE ε4 carriers and APOE non-ε4 carriers to explore the relationship between the functional connectivity of networks and cognitive function. These associations were significant if *p* was less than 0.05. All statistical analyses were performed using SPSS version 17.0 for Windows.

## 3. Results

### 3.1. Neuropsychological Characteristics of the Large Sample

Our study included 34 patients with APOE ε4 and 199 age-matched APOE ε4 non-carriers (Figure 1).

There were no significant differences in age, education or gender (*p* > 0.05). The APOE ε4 carriers group had worse executive function (Stroop C-B time, *p* = 0.048; TMT B time, *p* = 0.044), memory (Digit span-forward, *p* = 0.033) and language (CVFT, *p* = 0.032) than APOE ε4 non-carriers group (Table 1).

### 3.2. Selectively Altered RSNs in Patients with Hypertension

In the MRI study, 19 hypertensive patients carrying the APOE ε4, another 19 age-, education- and sex-matched non-carriers were included (Table 2). To investigate the pattern of the FC of hypertension patients, the best-fit components for the LFP, RFP, SN and DMN were obtained by the ICA group (*p* < 0.05, FWE corrected) (Figure 2). We performed ANCOVA tests on each of the four RSNs, contrasting the individual, back-reconstructed IC patterns of both groups, with age, gender, and education as covariates. The LFP network revealed a significant group difference (*p* < 0.05, FWE corrected). None of the ANCOVA tests on the other networks revealed a significant group difference (*p* < 0.05, FWE corrected). The following areas of the LFP network’ associated ICA pattern demonstrated decreased connectivity in the ε4-carrier group: left precentral (−51, 12, 36) and right triangle inferior frontal gyrus (54, 30, 24). The following area demonstrated increased connectivity in the ε4-carrier group: right angular gyrus (60, −54, 33) (Figure 3).

### 3.3. Frontal-Parietal Network Functional Connectivity Is Correlated with Behavior

The results showed that within the frontal-parietal network, significant FC of the left precentral gyrus × APOE ε4-status interaction effects were found in memory (Digit span-backward, *p* = 0.03, uncorrected), language (CVFT, *p* = 0.02, uncorrected) and processing speed (Stroop B time, *p* = 0.02, uncorrected) and significant FC of the right angular × APOE ε4-status interaction effects were found in memory (Digit span-backward, *p* = 0.02, uncorrected). Further analysis revealed that the FC of the left precentral gyrus was significantly correlated with memory (Digit span-backward, *r* = −0.50, *p* = 0.03), language (CVFT, *r* = −0.54, *p* = 0.02) and processing speed (Stroop B time, *r* = 0.53, *p* = 0.02) only in the ε4-carrier group. The FC of the right angular was significantly correlated with memory (Digit span-backward, *r* = 0.53, *p* = 0.02) only in the ε4-carrier group, and there was no significant correlation between the FC and other cognitive performance (*p* > 0.05). There was no significant correlation between the FC of the right inferior frontal gyrus and cognitive performance in each group (*p >* 0.05) (Figure 4).

Finally, our findings should be interpreted with caution because when we corrected for multiple comparisons (FDR) to group differences and group correlations, we found that not all the *p*-values could survive multiple comparisons corrections.

## 4. Discussion

The current study assessed the joint effects of hypertension and APOE ε4 risk genes on cognitive function in a large sample of elderly people. Cognitively, APOE ε4 carriers with hypertension mainly showed decreased execution functions, memory and language in the large cohort study. Only in the LFP networks were there obvious abnormal patterns of FC in APOE ε4 carrier hypertensive patients. The important finding was that the abnormal FC pattern in the LFP network was significantly related to the poorer performance in memory, language and processing speed. Our results suggested that APOE ε4 risk genes target a specific pattern of cognitive decline and FC changes and elevate frontoparietal dysfunction in hypertensive patients. These results may help us to understand the genetic mechanism by which hypertension serves as a potential risk factor for dementia and cognitive decline.

There are a few large-sample studies about the joint effect of hypertension and APOE-related cognitive decline, such as the Personality and Total Health (PATH) through Life project [25], the Honolulu-Asia Aging Study [26] and the Tone Project [19], and these studies have shown that hypertensive patients with APOE ε4 carriers can aggravate cognitive decline, especially in cognitive flexibility, working memory and episodic memory. Our results are consistent with those longitudinal studies, indicating that the APOE ε4 carrier patients showed decreased performance in several cognitive domains compared with the non-carriers, mainly in execution function. We also found that the APOE ε4 carrier group had a lower performance in language fluency and working memory tests, which might require more execution resources.

There are significant differences in the alterations of the LFP network between APOE ε4 carriers and APOE ε4 non-carriers under FWE correction. There was evidence showed that the FP network plays a vital role in executive function, attention control, and working-memory processing [27]. Moreover, the alterations in the frontal and parietal regions of the brain are highly correlated with cognitive deficits in hypertension [10,11,12]. The results reported here are in line with previous findings that a pattern of topologically worse connections focused on the frontoparietal network posterior parietal cortex in hypertensive patients [13], and the increased gene risk is accompanied by reduced functional brain activity in parietal and frontal areas [28,29], which means that APOE ε4 may aggravate frontal and parietal neurodegeneration patterns in hypertensive patients.

The significant regions included the left precentral gyrus, right IFG and angular regions. The left precentral gyrus was more susceptible to cognitive impairments. Reduced grey matter volume was also observed in precentral cortical regions in AD [30] and MCI [31]. The cortical precentral gyrus is also reported to be thinner in the APOE ε4 carriers than in non-ε4 allele carriers [32]. We found that decreased FC in the left precentral gyrus was related to worse memory, language and processing speed in APOE ε4 carriers. The IFG is known to participate in the maintenance of memory [33,34] and verbal fluency, which is negatively correlated with age [35,36] and is decreased in patients with AD [37]. Nevertheless, the inferior frontal regions play a critical role in protecting against the negative impact of neurodegeneration among people at risk for AD [36]. The present study found that altered connections in functional connectivity of IFG were not significantly correlated with cognitive function, which may indicate that hypertensive patients with APOE ε4 carriers could maintain normal cognitive function despite altered functional connectivity. The inferior parietal lobule, including the angular gyrus, is the core structure of the frontal parietal control system and also an important component of the default-mode network [38], whose functions involve cognitive control, memory extraction [39,40], perceptual information integration and conflict monitoring [38]. We found that increased FC of the angular regions was correlated with working memory. Actually, in the large sample study and the MRI study, we did not find a significant difference between APOE ε4 carriers and non-carriers in the working memory test—Digit span-backward. Previous studies have found no significant differences between the two groups in neuropsychological test performance in cognitively normal people. We could speculate that this cognitive ability may be impaired by hypertension indiscriminately, so there is no significant difference between the two groups. However, from the perspective of functional connectivity, functional connectivity abnormalities were significantly correlated with working memory. The current findings of significant differences and correlations could partly explain the effect of APOE ε4 on cognitive decline in hypertensive patients.

There are several explanations for the interaction between hypertension and the APOE genotype in relation to cognitive impairment pathology. First, the mouse model indicated the APOE ε4 genotype seems less able to adjust to a more defiant environment due to the impaired synaptic plasticity and delayed the astroglial repair process [41,42]. Second, APOE ε4 could increase the risk of cardiovascular disease [43], which is in turn associated with the pathology of AD. Third, APOEε4 could aggravate the effect of hypertension on tau levels [44].

Our study has several limitations. First, before patients were diagnosed with hypertension, their brains may have been affected by fluctuations in blood pressure. However, this effect cannot be assessed accurately prior to diagnosis. Second, the present study is cross-sectional in nature. Continued follow-up of this sample will help to further elucidate the neural mechanisms underlying the association between APOE and hypertension. Third, the sample in MRI is slightly small because the exclusion criteria and the ratio of APOE ε4 in Chinese elderly individuals is only approximately 15%, which may influence the results of cognition differences, functional connectivity and its relationship with cognition. Fourth, the present study did not set the non-hypertensive controls APOE ε4 carrier and non-carrier controls, it is because we only pay more attention to disease risk factors on the impact of the cognition and its neural mechanism, we did not test for risk genes in non-disease patients. Given the rigor of the research design, we would like to include corresponding participants and expand the sample size for subsequent studies to clarify possible confounding results.

We should note that we did not consider if there are differences in the type of hypertensive medication used by the different groups. Based on the existing research results, there is no significant difference in the effects of different drugs for hypertension. For example, researchers have found that there is no difference in cognition or brain function between two medications, a beta blocker (atenolol) and an angiotensin converting enzyme inhibitor (lisinopril) [45]. This may explain when researchers explored the effects of hypertension on cognitive and brain function in older adults, they also did not take their different medications into account [46]. In addition, in APOE ε4 carriers, six participants (31.6%) took angiotensin receptor blockers medications, eight participants (42.1%) took calcium channel blocker medications, one participant (5.2%) took β blockers medications, and four participants (21.1%) took compound antihypertensives medications. In APOE ε4 non-carriers, four participants (21.1%) took angiotensin receptor blockers medications, nine participants (47.4%) took calcium channel blocker medications, and six participants (31.5%) took compound antihypertensives medications. The result of the Chi-square test showed that there is no difference between group and medications, so we did not explore the effects of different antihypertensive medications on cognition and brain function. In fact, answers to this important question may be provided by performing specialized larger prospective studies to analyze the impact of medications on the identified microstructural and functional brain alterations.

## 5. Conclusions

In conclusion, our results suggest that APOE ε4 carriers are at increased risk for cognitive decline and abnormal FC in the left FP network if they suffer from hypertension as well. Our data imply that APOE ε4 expands the FC alteration, which is related to the cognitive impairment pattern in hypertensive patients. Possible clinical implications could be that clinicians should be more aware of hypertension in these APOE ε4 carriers. Furthermore, treatment trials for hypertension using dementia or cognitive decline as an outcome measure should stratify their results for the APOE genotype.

## Figures and Tables

**Figure 1 brainsci-12-00515-f001:**
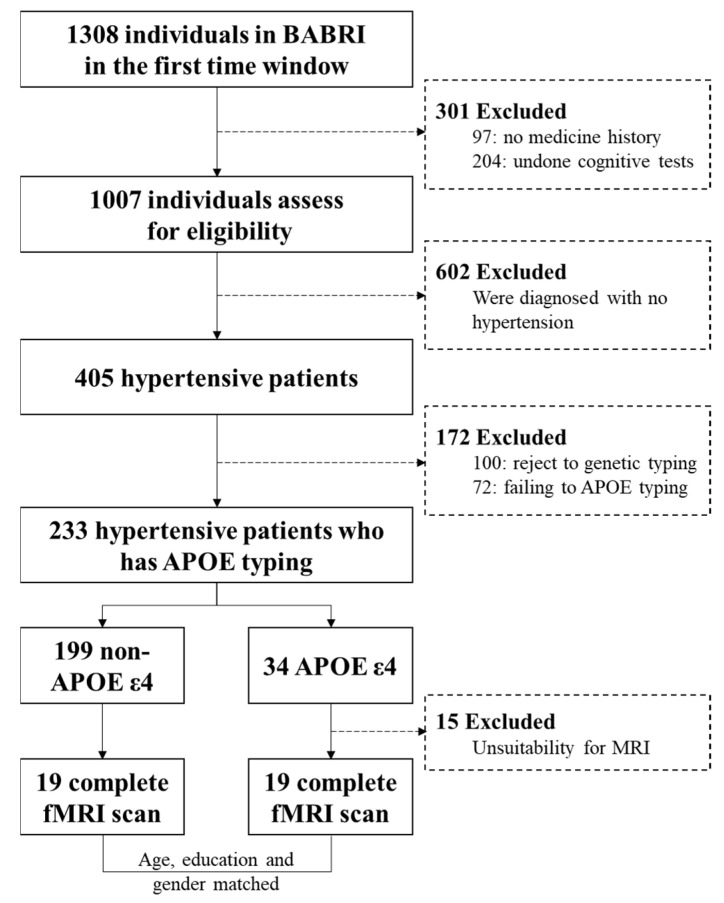
Participant flow chart. BABRI: Beijing Aging Brain Rejuvenation Initiative; APOE: Apolipoprotein E; fMRI: functional magnetic resonance imaging; MRI: magnetic resonance imaging.

**Figure 2 brainsci-12-00515-f002:**
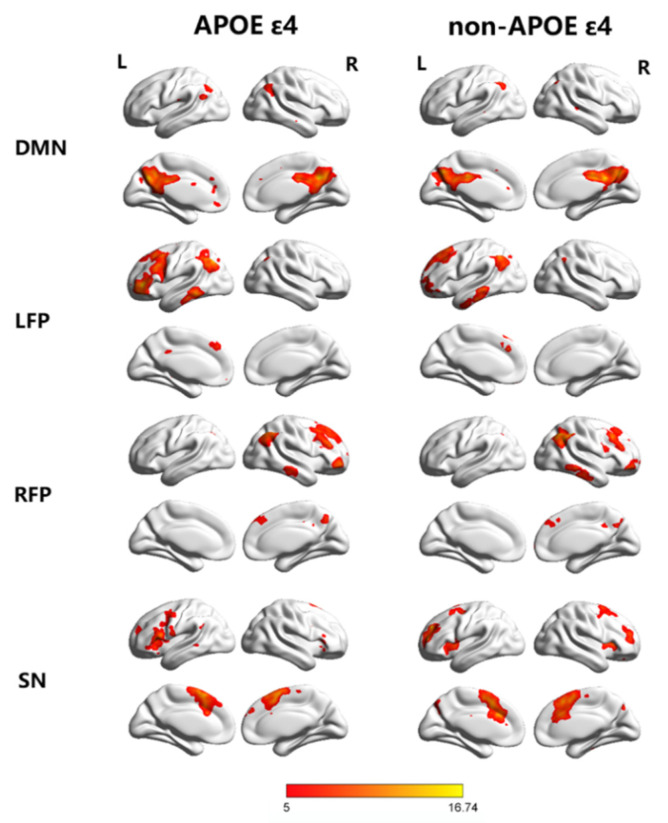
Group ICA estimated resting-state patterns grouped as the default mode network (DMN), left frontoparietal network (LFP), right frontoparietal network (RFP) and salience network (SN) in each group (Color-coded were t value, *p* < 0.05, Family Wise Error (FWE) corrected; L/R, left/right side).

**Figure 3 brainsci-12-00515-f003:**
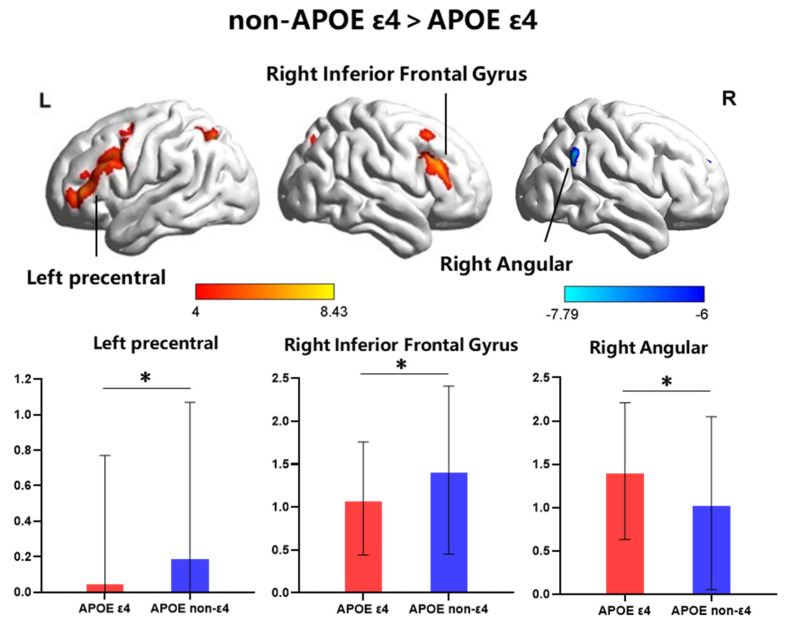
The significant group difference in functional connectivity within the LFP network. The *x*-axis represents groups and the *y*-axis represents the functional connectivity. In the left precentral and right triangle inferior frontal gyrus, the APOE ε4-carrier group showed decreased connectivity to the APOE non-ε4 group. However, in the right angular gyrus, the APOE ε4-carrier group showed increased connectivity than APOE non-ε4 group. *, *p* < 0.05, FWE corrected.

**Figure 4 brainsci-12-00515-f004:**
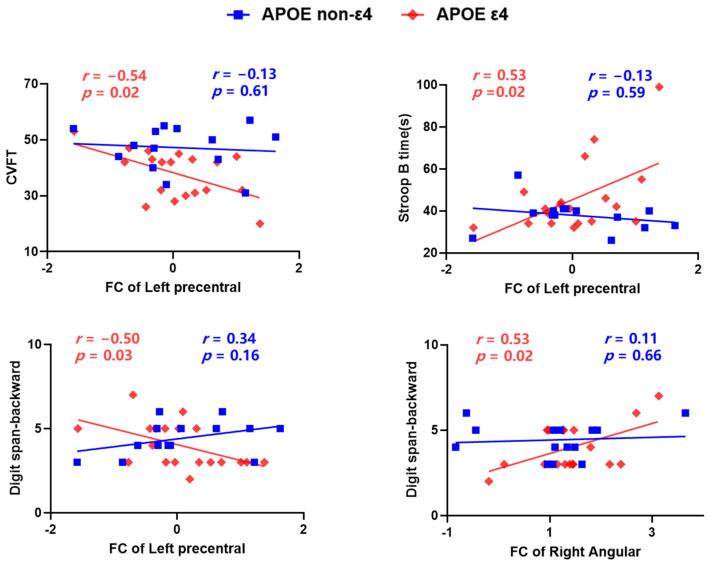
The relationship between functional connectivity within the frontal-parietal network and cognition. The functional connectivity (FC) of the left precentral regions is correlated with the performance of Digit span-backward, Category Verbal Fluency Test (CVFT) and the reaction times of the Stroop B test only in the ε4-carrier group. The functional connectivity of the right angular is correlated with the performance of digit span-backward only in the ε4-carrier group. *p* < 0.05, uncorrected.

**Table 1 brainsci-12-00515-t001:** Characteristics and neuropsychologic test results of participants in large sample from Beijing communities.

	APOE ε4 (*n* = 34)	Non-APOE ε4 (*n* = 199)	*F*	*p*
AGE (years)	66.12 ± 6.848	65.23 ± 7.265	0.66	0.509
SEX (male, %)	12, 35.3%	70, 35.2%	0.00	0.566
Education (years)	10.74 ± 3.387	11.11 ± 3.619	−0.57	0.570
MMSE	27.53 ± 1.674	27.87 ± 1.577	0.965	0.327
Memory				
AVLT	5.09 ± 2.598	5.77 ± 2.490	1.69	0.195
ROCF-delay	13.71 ± 5.745	13.85 ± 6.375	0.03	0.856
Digit span-forward	6.85 ± 1.329	7.39 ± 1.221	4.59	0.033
Digit span-backward	4.06 ± 1.349	4.37 ± 0.955	0.303	0.586
Executive function				
Stroop C-B time (s)	41.56 ± 14.724	37.04 ± 11.081	3.95	0.048
TMT B time (s)	184.09 ± 60.452	163.96 ± 46.934	4.11	0.044
Processing speed				
TMT A time (s)	62.26 ± 20.211	57.91 ± 20.236	0.69	0.407
Stroop B time (s)	38.74 ± 9.291	37.61 ± 7.815	0.36	0.547
Visuo-spatial ability				
ROCF-copy	33.18 ± 2.564	33.20 ± 3.378	0.01	0.907
CDT	24.36 ± 3.141	24.61 ± 3.490	0.02	0.882
Language				
CVFT	42.53 ± 8.140	45.83 ± 8.580	4.67	0.032
BNT	22.82 ± 4.196	23.41 ± 3.454	0.35	0.556

Notes: The measured data are represented by mean and standard deviation. The comparisons of age, education and neuropsychological assessment were performed with ANCOVA analyses (*F* values). The p-value for gender was obtained using a Chi-square test. Mini-Mental State Examination (MMSE); AVLT: Auditory Verbal Learning Test; ROCF: Rey-Osterrieth Complex Figure test; Stroop Test: Stroop Color and Word Test; TMT: Trail Making Test; CDT: Clock-Drawing Test; CVFT: Category Verbal Fluency Test; BNT: Boston Naming Test; *p* < 0.05, uncorrected, the same below.

**Table 2 brainsci-12-00515-t002:** Characteristics and neuropsychologic test results of participants in MRI sample.

	APOE ε4 (*n* = 19)	Non-APOE ε4 (*n* = 19)	*F*	*p*
AGE (years)	68.26 ± 6.94	68.16 ± 4.07	−0.06	0.96
SEX (male, %)	11, 47.4%	9, 57.9%	0.11	0.745
EDU (years)	10.13 ± 3.01	11.74 ± 3.48	1.52	0.14
MMSE	26.58 ± 4.22	27.89 ± 1.66	1.42	0.24
Memory				
AVLT	4.53 ± 3.29	5.21 ± 2.12	0.08	0.78
ROCF-delay	12.26 ± 6.23	14.11 ± 5.91	0.83	0.37
Digit span-forward	6.89 ± 1.33	8.26 ± 1.76	5.42	0.026
Digit span-backward	4.00 ± 1.33	4.37 ± 0.96	0.43	0.52
Executive function				
Stroop C-B time (s)	43.26 ± 13.78	31.94 ± 13.41	5.02	0.033
TMT B time (s)	204.28 ± 62.48	167.42 ± 55.74	3.06	0.09
Processing speed				
TMT A time (s)	70.05 ± 33.30	59.53 ± 15.94	1.16	0.29
Stroop B time (s)	46.05 ± 17.12	38.74 ± 6.72	2.42	0.13
Visuo-spatial ability				
ROCF-copy	31.84 ± 3.75	34.21 ± 1.99	4.37	0.044
CDT	23.32 ± 4.74	24.63 ± 3.62	0.57	0.46
Language				
CVFT	37.89 ± 8.69	45.32 ± 8.33	4.97	0.033
BNT	21.95 ± 4.31	23.58 ± 3.42	3.58	0.07

Notes: MRI sample means participants in Magnetic resonance imaging (MRI) Studies.

## Data Availability

The data presented in this study are available on request from the corresponding author. The data are not publicly available due to privacy and ethical restrictions.

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
