# Peer review of "The Effect of APOE ɛ4 on the Functional Connectivity in Frontoparietal Network in Hypertensive Patients"

_brainsci, 2022, doi:10.3390/brainsci12050515_

Round 1

Reviewer 1 Report

The study discusses the general correlations between cognitive decline and changed functional connectivity in specific brain networks in hypertensive patients when they are also APOEε4 carriers.

I do have a couple of concerns, mostly regarding clarity of presentation.

  1. Figure 4 has several correlations between FC and different cognitive measures. The authors merely state that these correlations exist, without talking about what these correlations mean in broader context. For example, in Figure 4A, left and middle panel show positive correlations in the left and middle panel, and negative correlation in the right most panel. What do these differences in the sign of the correlations mean? If I am understanding correctly, aren't those opposite effects? I think the reader would like some more clarity on this.
  2. In lines 216-218, the authors cite previous studies about APOEε4 and hypertension. It would be useful to get a brief summary of what those studies found and how results of current study add/contrast/complement those results.
  3. Not a comment, but I appreciate the summary of limitations of the study on lines 255-283.

Author Response

The study discusses the general correlations between cognitive decline and changed functional connectivity in specific brain networks in hypertensive patients when they are also APOEε4 carriers. I do have a couple of concerns, mostly regarding clarity of presentation.

Point 1: Figure 4 has several correlations between FC and different cognitive measures. The authors merely state that these correlations exist, without talking about what these correlations mean in broader context. For example, in Figure 4A, left and middle panel show positive correlations in the left and middle panel, and negative correlation in the right most panel. What do these differences in the sign of the correlations mean? If I am understanding correctly, aren't those opposite effects? I think the reader would like some more clarity on this.

Response 1: Thank you for your comments about our description of Method and Result.

The purpose of correlation analysis is to explore the correlation between the functional connectivity of networks and neuropsychological performance. In order to describe the tests that measure different cognitive domains more clearly, we have added the relevant content in 2.1.3 in line 134-138 (display all marked page numbers in revision mode, the same below). It can be seen that the tests measuring processing speed and executive function used the time (also see table 1 and 2) when the participants completed the tests, however others used their actual scores, so the correlation between cognitive ability and FC was opposite to other tests. In addition, we were not only described the correlations between FC and cognition, we also discussed them in conjunction with other imaging results, as shown in the fourth paragraph of the Discussion, please see the revised edition manuscript in the attachment.

Point 2: In lines 216-218, the authors cite previous studies about APOEε4 and hypertension. It would be useful to get a brief summary of what those studies found and how results of current study add/contrast/complement those results.

Response 2: Thank you for your helpful suggestion about the relevant descriptions of the Discussion.

In the second paragraph of the discussion, we just cite previous studies about APOE-ε4 and hypertension. To better illustrate the relationship between our results and their findings, we have added a brief summary of what those studies found and how results of current study contrast those results, please see line 288-295.

Point 3: Not a comment, but I appreciate the summary of limitations of the study on lines 255-283.

Response 3: Thank you for your recognition of this part in the Discussion.

Hypertension as a chronic disease, there must be researchers or readers pay attention to the effects of cognitive and FC caused by medication, so we described the type of medication these patients was taking. Unfortunately, due to the particularity of genes, our participants are relatively small, so we cannot explore the effects of different treatments for the time being in the results.

Finally, in order to improve the English language of our manuscript, we also invited experts to read it for language polish.

Reviewer 2 Report

Summary:

The goal of this study was to determine the interactions between the presence of APOE ε4 allele and a diagnosis of hypertension on cognitive dysfunction and disrupted resting state functional connectivity. The hypothesis is well define, the study design is appropriate, and the manuscript is very well written. The content of the manuscript is well organized with optimal information flow. The manuscript contains an appropriate number of tables and figures.

A couple of concerns stand out. These should be explained or addressed as limitations: 1) how weak the group differences are in terms of cognition differences. The premise of the study is increased cognitive decline in APOE ε4 carriers compared to non-carriers, but the data suggests that the decline may be minimal. 2) the lack of objective controls in the design, which leaves the author and reader to assume the non-carrier hypertensives have healthier FC patterns than carrier hypertensives. Using the hypothesis to explain the result introduces a circularity which has not been addressed or noted in the limitations.

Comments:

  • Introduction: Functional connectivity is not described in the Introduction before the abbreviated “FC” is used. This statement is very misleading: “The FC of RSNs is based on the white matter (WM) tract linkage and facilitates the ongoing interregional neuronal communication”
    • Please introduce BOLD rsfMRI and FC in the introduction. The association with WM tracts is not essential in the introduction. There have been studies exploring the effects of APOE ε4 on rsfMRI and those which explore Hypertension effects of rsfMRI (albeit independently). It might be sufficient to cite them for background information and study motivation. Should the authors choose to retain WM information – please rewrite the introduction to clarify the misleading statement highlighted above.
  • Methods: Please provide more detail on the neuropsychological battery. The tests mentioned in the results should be named/detailed in the Methods.
  • MRI methods: This statement is confusing, please rephrase: “Because there are only 19 hypertensive patients carrying the APOE ε4, we chose 19 age-, education- and sex-matched non-carriers (Fig.1) from the 233 total subjects to MRI scans.” The fact that only 19 hypertensive APOE ε4 patients could be scanned becomes clear only later when looking at the sample breakdown in the Results.
  • fMRI acquisition: How many time points were collected? Or rather, how long were the rsfMRI scans. The matrix size seems very small – what were the in-plane voxel sizes and the field of view?
  • MRI acquisition and data analysis: No details on structural MRI are provided. Were anatomical T1-w scans not acquried and used for spatial normalization to standard MNI template? With the rsfMRI matrix size being 64x64 to cover the entire brain, let’s assume a 200m field of view, this would result in 3.25x3.25mm2 in-plane resolution. The gray matter – white matter contrast with such large voxel sizes would be insufficient to perform accurate spatial normalization to the template space.
  • Statistical analyses: Please rephrase “To neuropsychological assessment, a covariance analysis was used to compare” to perhaps “Covariance analyses were used to compare neuropsychological assessments”.
  • Statistical analyses: Why were Pearson’s correlation analyses performed separately for ε4 carriers and non-carriers? Given the study goal was to explore the interaction, why not study them in a single model with interaction between functional connectivity and carrier-status as a categorical variable?
  • Results: Table 1: Were the various group differences and group correlations performed on SPSS corrected for multiple comparisons? There are close to 12 neuropsychological tests, and some of the p values are very close to 0.05. Could the authors please make a note on if the p-values in the tables survive multiple comparisons corrections (false discovery rate may suffice, Bonferroni correction may be too strict) Based on a quick look, most group differences would not survive even FDR correction.
  • Results: Figure 3, please label the y-axis
  • Discussion: Once again, the use of WM degeneration related references to support functional connectivity results is inappropriate. “The results reported here are in accord with previous findings that older APOE .4 carriers show increased frontal white matter degradation[29-30]. APOE .4 aggravated frontal and parietal neurodegeneration patterns in hypertensive patients.” The work in this paper does not show any markers of degradation or neurodegeneration.
  • Discussion: “The cortical precentral gyrus is also reported to be thinner”. Please rephrase the citation.
  • One of the biggest limitations in this study is the lack of non-hypertensive controls – carriers and non-carriers. The reason this is an issue is because the authors report decreased connectivity in 2 regions and increased connectivity in 1 region for APOE ε4 carriers. While this may be simply labelled as “abnormal”, what makes the hypertensive-non-carrier hypoactivation of angular gyrus “normal”?
  • Correlation does not mean causation, “abnormal functional connectivity of angular gyrus may be an important cause of memory loss in APOE” is not supported by the data. In fact, digit-span backward shows no significant difference between carriers and non-carriers in the whole study sample or in the MRI subset.
  • The authors mention the lack of knowledge about blood pressure fluctuations pre-diagnosis. Do the authors have data on disease duration? Or treatment duration? This is optionally helpful to dissect some of these relationships.

Author Response

Point 1: Introduction

1)Functional connectivity is not described in the Introduction before the abbreviated “FC” is used. This statement is very misleading: “The FC of RSNs is based on the white matter (WM) tract linkage and facilitates the ongoing interregional neuronal communication”

2)Please introduce BOLD rsfMRI and FC in the introduction. The association with WM tracts is not essential in the introduction. There have been studies exploring the effects of APOE ε4 on rsfMRI and those which explore Hypertension effects of rsfMRI (albeit independently). It might be sufficient to cite them for background information and study motivation. Should the authors choose to retain WM information – please rewrite the introduction to clarify the misleading statement highlighted above.

Response 1: Thank you for your helpful comments about the description of the Introduction.

As far as our subject is concerned, the association with WM tracts seemed not essential in the introduction, however, anatomically, the white matter (WM) tracts are considered as the anatomic links of RSNs and densely interconnect the regions within these RSNs. Interestingly, both hypertension and APOE ε4 influence the WM tracts, and there is also a significantly higher subcortical white matter lesion volume in APOE ε4 carriers with hypertension than in non-ε4 carriers, therefore, there may be the joint effect of the APOE risk gene and hypertension on the functional connectivity (FC) of RSNs, so we retain some WM information. In addition, we have also added the description of studies exploring the effects of APOE ε4 on rsfMRI and the effects of hypertension on rsfMRI, so the Introduction has been significantly revised, including before the abbreviated word is used, we described it in its full name, please see the whole Introduction in the revised edition manuscript in the attachment. 

Point 2: Methods: Please provide more detail on the neuropsychological battery. The tests mentioned in the results should be named/detailed in the Methods.

Response 2: Thank you for your helpful and constructive comments about the description of the Methods.

In order to describe the tests that measure different cognitive domains more clearly, we have added the relevant content in 2.1.3 in line 134-138 (display all marked page numbers in revision mode, the same below).

Point 3: MRI methods: This statement is confusing, please rephrase: “Because there are only 19 hypertensive patients carrying the APOE ε4, we chose 19 age-, education- and sex-matched non-carriers (Fig.1) from the 233 total subjects to MRI scans.” The fact that only 19 hypertensive APOE ε4 patients could be scanned becomes clear only later when looking at the sample breakdown in the Results.

Response 3: We have delated the confusing statement about the participants in MRI methods in line 145-147, and rephrased the related content in line 151-153.

Point 4: fMRI acquisition: How many time points were collected? Or rather, how long were the rsfMRI scans. The matrix size seems very small – what were the in-plane voxel sizes and the field of view?

Response 4: Sorry for not making clear explained about the parameters that related to MRI data acquisition, we have added the parameters to the manuscript, please see line 156-161. Acquisition in the resting state lasted for 8 min 5 s, and 240 image volumes were obtained. Field of view (FOV) =200×200 mm2, matrix=64×64, and the in-plane voxel sizes=3.125 mm.

Point 5: MRI acquisition and data analysis: No details on structural MRI are provided. Were anatomical T1-w scans not acquried and used for spatial normalization to standard MNI template? With the rsfMRI matrix size being 64x64 to cover the entire brain, let’s assume a 200m field of view, this would result in 3.25x3.25mm2 in-plane resolution. The gray matter – white matter contrast with such large voxel sizes would be insufficient to perform accurate spatial normalization to the template space.

Response 5: The details about structural MRI were as follow: T1-weighted, sagittal 3D magnetization-prepared rapid gradient echo sequences were acquired and covered the entire brain [176 sagittal slices, repetition time (TR)=1900 ms, echo time (TE)=3.44 ms, slice thickness=1 mm, flip angle=9°, inversion time=900 ms, field of view=256×256 mm2, acquisition matrix=256×256]. Actually, we used a standard brain template in the Montreal Neurological Institute coordinate space for spatial normalization rather than T1-weighted scans, so in the method, we did not show the details parameters about the structural MRI.

Point 6: Statistical analyses: Please rephrase “To neuropsychological assessment, a covariance analysis was used to compare” to perhaps “Covariance analyses were used to compare neuropsychological assessments”.

Response 6: We have rephrased the statement to “ANCOVA analyses were used to compare neuropsychological assessments between the two groups, with age, education, gender, and medical history as covariates.” in line 183-184.

Point 7: Statistical analyses: Why were Pearson’s correlation analyses performed separately for ε4 carriers and non-carriers? Given the study goal was to explore the interaction, why not study them in a single model with interaction between functional connectivity and carrier-status as a categorical variable?

Response 7: Thank you very much for your constructive comments and suggestions about the statistical analyses. According to your suggestion, we studied them in a single model with interaction between functional connectivity and APOE ε4 carriers-status as a categorical variable, and added the description in methods and result to the manuscript, please see line 187-192 and line 244-249.

Point 8: Results: Table 1: Were the various group differences and group correlations performed on SPSS corrected for multiple comparisons? There are close to 12 neuropsychological tests, and some of the p values are very close to 0.05. Could the authors please make a note on if the p-values in the tables survive multiple comparisons corrections (false discovery rate may suffice, Bonferroni correction may be too strict) Based on a quick look, most group differences would not survive even FDR correction.

Response 8: Thank you for your helpful and constructive comments on the Results.

According to your suggestion, we corrected for multiple comparisons (FDR) to group differences and group correlations, then we found that all the p-values couldn’t survive multiple comparisons corrections, so we added the ‘uncorrected’ to the results in line 215, line 244-248, and line 264-265.

Point 9: Results: Figure 3, please label the y-axis

Response 9: We have labeled the y-axis about Figure 3 in the legend, please see line 238-239.

Point 10: Discussion: Once again, the use of WM degeneration related references to support functional connectivity results is inappropriate. “The results reported here are in accord with previous findings that older APOE .4 carriers show increased frontal white matter degradation[29-30]. APOE .4 aggravated frontal and parietal neurodegeneration patterns in hypertensive patients.” The work in this paper does not show any markers of degradation or neurodegeneration.

Response 10: Thank you for your helpful and constructive comments on the Discussion.

It is do inappropriate to use the WM degeneration related references to support functional connectivity results, combined with the relevant description in the introduction, we have added the resting state functional connection related reference to discuss the results of the present study, please see line 295-300.

Point 11: Discussion“The cortical precentral gyrus is also reported to be thinner”. Please rephrase the citation.

Response 11: We have rephrased “The cortical precentral gyrus is also thinner in the APOE ε4 carriers than in non-ε4 allele carriers” to “The cortical precentral gyrus is also reported to be thinner in the APOE ε4 carriers than in non-ε4 allele carriers”, please see line 306.

Point 12: One of the biggest limitations in this study is the lack of non-hypertensive controls – carriers and non-carriers. The reason this is an issue is because the authors report decreased connectivity in 2 regions and increased connectivity in 1 region for APOE ε4 carriers. While this may be simply labelled as “abnormal”, what makes the hypertensive-non-carrier hypoactivation of angular gyrus “normal”?

Response 12: Thank you for your constructive comments. As you mentioned, the present study did not set the non-hypertensive controls APOE ε4 carrier and non-carrier controls, it is because we only pay more attention to disease risk factors on the impact of the cognition and its neural mechanism, we did not test for risk genes in non-disease patients. Given the rigor of the research design, we would like to include corresponding participants for subsequent studies to clarify possible confounding results. The similar descriptions have added to the limitations, please see line 348-353.

Point 13: Correlation does not mean causation, “abnormal functional connectivity of angular gyrus may be an important cause of memory loss in APOE” is not supported by the data. In fact, digit-span backward shows no significant difference between carriers and non-carriers in the whole study sample or in the MRI subset.

Response 13: As for the discussion on the result of functional connectivity and cognition, the original manuscript was not sufficient, so we rewrite corresponding descriptions included the inaccurate description you mentioned, please see line 312-316 (the fourth paragraph of the Discussion).

Point 14: The authors mention the lack of knowledge about blood pressure fluctuations pre-diagnosis. Do the authors have data on disease duration? Or treatment duration? This is optionally helpful to dissect some of these relationships.

Response 14: Acturally, we couldn’t assess their brain functions accurately prior to hypertension diagnosis, and we once asked patients about the disease duration, but unfortunately the duration they reported was only based on their own estimates, and we didn't consider the relevant data for accuracy.

Point 15: A couple of concerns stand out. These should be explained or addressed as limitations:

1) how weak the group differences are in terms of cognition differences. The premise of the study is increased cognitive decline in APOE ε4 carriers compared to non-carriers, but the data suggests that the decline may be minimal.

2) the lack of objective controls in the design, which leaves the author and reader to assume the non-carrier hypertensives have healthier FC patterns than carrier hypertensives. Using the hypothesis to explain the result introduces a circularity which has not been addressed or noted in the limitations.

Response 15: Thank you for your helpful comments again.

Due to the weak the group cognition differences, we added some description in limitation. Furthermore, the second concern is same as point 11, the similar descriptions have added to the limitations in line 348-353, and we would like to include corresponding participants for subsequent studies to clarify possible confounding results.

Finally, in order to improve the English language of our manuscript, we also invited experts to read it for language polish.

Reviewer 3 Report

The authors examined the differences in cognitive and brain function between hyperintensive patients with APOEε4 carriers and non-carriers. The study is interesting but has several flaws like serious writing issues (extensive editing of English laguage is required) in such a point that it is really difficult in some points to follow. 

Introduction

It should be extented and make a more clear hypothesis about the study. Quality and clarity of writing is poor. Extensive editing of English language is required. There were a lot of parts that I could not understand. Examples: Lines 43-45, 47 etc...

Methods

Please give more details about RS sequence (number of volumes, duration), line 114-116.

Please explain why  you chose in the ICA analysis 25 components (It is related with the duration of the RS sequence), line 129

Please clarify if the correlations between functional connectivity and neuropsychological performance have been corrected for multiple comparisons

Lines 133-135: The identification of RSNs was based on visual inspection. How bias do you think is this procedure? Could you think alternatives methods will help? please see Demertzi et al. 2014.

Results

Please improve quality of Figure 1

Figure 4: Insignificant correlation scatter plots could be remove

Discussion

As the introduction, extensive editing of English writing is needed. Some examples: Lines 222-224, 226-229, 237-241 etc...

Author Response

The authors examined the differences in cognitive and brain function between hyperintensive patients with APOEε4 carriers and non-carriers. The study is interesting but has several flaws like serious writing issues (extensive editing of English laguage is required) in such a point that it is really difficult in some points to follow. 

Point 1: Introduction

It should be extented and make a more clear hypothesis about the study. Quality and clarity of writing is poor. Extensive editing of English language is required. There were a lot of parts that I could not understand. Examples: Lines 43-45, 47 etc...

Response 1: Thank you for your helpful comments and suggestions on the Introduction.

To better fit the research topic, we have added the description of studies exploring the effects of APOE ε4 on rsfMRI and the effects of hypertension on rsfMRI, and we also clarified the hypothesis. We have revised the Introduction and polished the language by experts, including before the abbreviated word is used, we described it in its full name, please see the whole Introduction in the changed manuscript in the attachment.

Point 2: Methods

Please give more details about RS sequence (number of volumes, duration), line 114-116.

Response 2: Sorry for not making clear explained about the parameters that related to MRI data acquisition, we have added the parameters to the manuscript, please see line 156-161 (display all marked page numbers in revision mode, the same below). Acquisition in the resting state lasted for 8 min 5 s, and 240 image volumes were obtained.

Point 3: Methods

Please explain why you chose in the ICA analysis 25 components (It is related with the duration of the RS sequence), line 129

Lines 133-135: The identification of RSNs was based on visual inspection. How bias do you think is this procedure? Could you think alternatives methods will help? please see Demertzi et al. 2014.

Response 3: Thank you for your helpful comments and suggestions about the details of the Methods. Based on previous studies [1-2], we have ever set 20, 25, 30 and even more components, respectively, then we found that only 25 was the best one, with fewer components may mix multiple networks, and more components seem to divide those networks into smaller ones. In addition, these networks that selected by visual inspection were done by experienced researchers. We have ever permitted similarity tests to the DMN and SN templates in the GIFT software, and found that they were reliable. The FP network contained the major brain regions, and we have ever compared it with the network from previous studies [3]. Therefore, we think that the networks that we've chosen should be fine. In spite of this, for the consideration of data rigor, we are willing to try the method you mentioned that used another control group [4] in the follow-up study, and permit a spatial similarity test with the selected networks to ensure the reliability of network components. Thank you again for your constructive suggestions.

Point 4: Methods

Please clarify if the correlations between functional connectivity and neuropsychological performance have been corrected for multiple comparisons

Response 4: According to your suggestion, we corrected for multiple comparisons (FDR) to group differences and group correlations, then we found that all the p-values couldn’t survive multiple comparisons corrections, so we added the ‘uncorrected’ to the results in line 215, line 244-248, and line 264-265.

Point 5: Results

Please improve quality of Figure 1

Figure 5: Insignificant correlation scatter plots could be remove

Response 5: We have redrawn Figure 1, and moved the insignificant correlation in Figure 4 and its legend in line 260-265 (display all marked page numbers in revision mode, the same below).

Point 6: Discussion

As the introduction, extensive editing of English writing is needed. Some examples: Lines 222-224, 226-229, 237-241 etc...

Response 6: In order to improve the English language of our manuscript, we also invited experts to read the Discussion for language polish, please see the whole Discussion.

Finally, in order to improve the English language of our manuscript, we also invited experts to read it for language polish.

Reference

  1. Li X; Liang Y; Chen Y; Zhang J; Wei D; Chen K; et al. Disrupted Frontoparietal Network Mediates White Matter Structure Dysfunction Associated with Cognitive Decline in Hypertension Patients. Journal of Neuroscience the Official Journal of the Society for Neuroscience 2015, 35, 10015-10024.
  2. Smith SM; Fox PT; Miller KL; Glahn DC; Fox PM; Mackay CE; et al. Correspondence of the brain's functional architecture during activation and rest. Proceedings of the National Academy of Sciences 2009, 106, 13040-13045.
  3. Cole D; Smith S; & Beckmann C. Advances and Pitfalls in the Analysis and Interpretation of Resting-State FMRI Data. Frontiers in Systems Neuroscience 2010, 4, 1-15.
  4. Demertzi A; Soddu A; Faymonville ME; Bahri MA; Gosseries O; Vanhaudenhuyse A; et al. Hypnotic modulation of resting state fMRI default mode and extrinsic network connectivity. Prog Brain Res 2011, 193, 309-322.

Reviewer 4 Report

The manuscript is well written and well planned research output. The authors provided enough experimental evidences to prove their hypothesis and the research design was outstanding. 

I would suggest to improve the quality of figures (Figure 1, 2, 4). Please enlarge them and improve the text. 

Please mention the numbers in each group in figure legends where applicable. 

In tables, please mark the statistically significant data in an appropriate way for better illustrations. Same suggestion applies to figure 4. 

If possible, could include a graphical abstract for better presentation of the manuscript summary. 

Author Response

Point 1: I would suggest to improve the quality of figures (Figure 1, 2, 4). Please enlarge them and improve the text.

Response 1: Thank you for your constructive comments and suggestions.

We have redrawn Figure 1, improved the Figure 2 and Figure 3, moved the insignificant correlation scatter plots in Figure 4 and bolded the significantly different results in Table 1 and Table 2, please see the changed manuscript in the attachment.

Point 2: In tables, please mark the statistically significant data in an appropriate way for better illustrations. Same suggestion applies to figure 4.

Response 2: Sorry for making a mistake about the number of the participants in MRI sample, we corrected it in line 229 (display all marked page numbers in revision mode, the same below). As there were 19 participants in both groups in MRI sample, we didn't indicate separately in figure legends.

Point 3: If possible, could include a graphical abstract for better presentation of the manuscript summary.

Response 3: Thank you again for your constructive suggestion, we try to graphical abstract for better presentation of the manuscript summary based on current research. However, we are not sure where to put it properly, so we put it at the line 523 in the revised manuscript for later decision, please see the revised edition manuscript in the attachment.

Finally, in order to improve the English language of our manuscript, we also invited experts to read it for language polish.

Round 2

Reviewer 2 Report

No further revisions necessary, the authors have addressed all my concerns from the first review.

Reviewer 3 Report

Thank you very much for addressing most of my comments. I would suggest to add in the results section 3.3 : "that our findings should interpret with caution because when we corrected for multiple comparisons (FDR) to group differences and group correlations, we found that all the p-values couldn’t survive multiple comparisons corrections"

Minor comments: Lines 324-325 Please replace "Considering 
that previous researchers" with "Previous studies have shown"

Author Response

Point 1: Thank you very much for addressing most of my comments. I would suggest to add in the results section 3.3 : "that our findings should interpret with caution because when we corrected for multiple comparisons (FDR) to group differences and group correlations, we found that all the p-values couldn’t survive multiple comparisons corrections"

Response 1: Thank you again for your constructive comments.

According to your suggestion, we have added the description about FDR correction at the end of result section 3.3 in line 258-260——“Finally, our findings should interpret with caution because when we corrected for multiple comparisons (FDR) to group differences and group correlations, we found that all the p-values couldn’t survive multiple comparisons corrections.”

Point 2: Minor comments: Lines 324-325 Please replace "Considering that previous researchers" with "Previous studies have shown"

Response 2: Thank you for your comments. We have replaced "Considering that previous researchers have found " into "Previous studies have found" in the new line 326-327.
